# BOOSTING THE ACTOR WITH DUAL CRITIC

**Bo Dai**[*1]**, Albert Shaw**[*1]**, Niao He**[2]**, Lihong Li**[3]**, Le Song**[1, 4]

[1] Georgia Institute of Technology, [2] University of Illinois at Urbana-Champaign
[3] Google AI, [4] Ant Financial Services Group
[1] `{bodai, ashaw596}@gatech.edu, lsong@cc.gatech.edu`
[2] `niaohe@illinois.edu,` [3] `lihongli.cs@gmail.com`

## ABSTRACT

This paper proposes a new actor-critic-style algorithm called Dual Actor-Critic or Dual-AC. It is derived in a principled way from the Lagrangian dual form of the Bellman optimality equation, which can be viewed as a two-player game between the actor and a critic-like function, which is named as dual critic. Compared to its actor-critic relatives, Dual-AC has the desired property that the actor and dual critic are updated *cooperatively* to optimize the same objective function, providing a more transparent way for learning the critic that is directly related to the objective function of the actor. We then provide a concrete algorithm that can effectively solve the minimax optimization problem, using techniques of multi-step bootstrapping, path regularization, and stochastic dual ascent algorithm. We demonstrate that the proposed algorithm achieves state-of-the-art performance across several benchmarks.

## 1 INTRODUCTION

Reinforcement learning (RL) algorithms aim to learn a policy that maximizes the long-term return by sequentially interacting with an unknown environment. Value-function-based algorithms first approximate the optimal value function, which can then be used to derive a good policy. These methods (Sutton, 1988; Watkins, 1989) often take advantage of the Bellman equation and use boot-strapping to make learning more sample efficient than Monte Carlo estimation (Sutton & Barto, 1998). However, the relation between the quality of the learned value function and the quality of the derived policy is fairly weak (Bertsekas & Tsitsiklis, 1996). Policy-search-based algorithms such as REINFORCE (Williams, 1992) and others (Kakade, 2002; Schulman et al., 2015a), on the other hand, assume a fixed space of parameterized policies and search for the optimal policy parameter based on unbiased Monte Carlo estimates. The parameters are often updated incrementally along stochastic directions that on average are guaranteed to increase the policy quality. Unfortunately, they often have a greater variance that results in a higher sample complexity.

Actor-critic methods combine the benefits of these two classes, and have proved successful in a number of challenging problems such as robotics (Deisenroth et al., 2013), meta-learning (Bello et al., 2016), and games (Mnih et al., 2016). An actor-critic algorithm has two components: the actor (policy) and the critic (value function). As in policy-search methods, actor is updated towards the direction of policy improvement. However, the update directions are computed with the help of the critic, which can be more efficiently learned as in value-function-based methods (Sutton et al., 1999; Konda & Tsitsiklis, 2003; Peters et al., 2005; Bhatnagar et al., 2009; Schulman et al., 2015b). Although the use of a critic may introduce bias in learning the actor, its reduces variance and thus the sample complexity as well, compared to pure policy-search algorithms.

While the use of a critic is important for the efficiency of actor-critic algorithms, it is not entirely clear how the critic should be optimized to facilitate improvement of the actor. For some parametric family of policies, it is known that a certain compatibility condition ensures the actor parameter update is an unbiased estimate of the true policy gradient (Sutton et al., 1999). In practice, temporal-difference methods are perhaps the most popular choice to learn the critic, especially when nonlinear function approximation is used (e.g., Schulman et al. (2015b)).

---

*Both authors equally contributed to the paper.

In this paper, we propose a new actor-critic-style algorithm where the actor and the critic-like function, which we named as dual critic, are trained *cooperatively* to optimize the same objective function. The algorithm, called *Dual Actor-Critic* , is derived in a principled way by solving a dual form of the Bellman equation (Bertsekas & Tsitsiklis, 1996). The algorithm can be viewed as a two-player game between the actor and the dual critic, and in principle can be solved by standard optimization algorithms like stochastic gradient descent (Section 2). We emphasize the dual critic is not fitting the value function for *current policy*, but that of the *optimal policy*. We then show that, when function approximation is used, direct application of standard optimization techniques can result in instability in training, because of the lack of convex-concavity in the objective function (Section 3). Inspired by the augmented Lagrangian method (Luenberger & Ye, 2015; Boyd et al., 2010), we propose *path regularization* for enhanced numerical stability. We also generalize the two-player game formulation to the multi-step case to yield a better bias/variance tradeoff. The full algorithm is derived and described in Section 4, and is compared to existing algorithms in Section 5. Finally, our algorithm is evaluated on several locomotion tasks in the MuJoCo benchmark (Todorov et al., 2012), and compares favorably to state-of-the-art algorithms across the board.

**Notation.** We denote a discounted MDP by $\mathcal{M} = (\mathcal{S}, \mathcal{A}, P, R, \gamma)$, where $\mathcal{S}$ is the state space, $\mathcal{A}$ the action space, $P(\cdot|s, a)$ the transition probability kernel defining the distribution over next-state upon taking action $a$ in state $x$, $R(s, a)$ the corresponding immediate rewards, and $\gamma \in (0, 1)$ the discount factor. If there is no ambiguity, we will use $\sum_a f(a)$ and $\int f(a) da$ interchangeably.

## 2 DUALITY OF BELLMAN OPTIMALITY EQUATION

In this section, we first describe the linear programming formula of the Bellman optimality equation (Bertsekas, 1995; Puterman, 2014), paving the path for a duality view of reinforcement learning via Lagrangian duality. In the main text, we focus on MDPs with finite state and action spaces for simplicity of exposition. We extend the duality view to continuous state and action spaces in Appendix A.2.

Given an initial state distribution $\mu(s)$, the reinforcement learning problem aims to find a policy $\pi(\cdot|s) : \mathcal{S} \to \mathcal{P}(\mathcal{A})$ that maximizes the total expected discounted reward with $\mathcal{P}(\mathcal{A})$ denoting all the probability measures over $\mathcal{A}$, *i.e.*,

$$\mathbb{E}_{s_0 \sim \mu(s)} \mathbb{E}_\pi \left[ \sum_{i=0}^\infty \gamma^i R(s_i, a_i) \right], \tag{1}$$

where $s_{i+1} \sim P(\cdot|s_i, a_i)$, $a_i \sim \pi(\cdot|s_i)$.

Define $V^*(s) := \max_{\pi \in \mathcal{P}(\mathcal{A})} \mathbb{E} \left[ \sum_{i=0}^\infty \gamma^i R(s_i, a_i)|s_0 = s \right]$, the Bellman optimality equation states that:

$$V^*(s) = (\mathcal{T}V^*)(s) := \max_{a \in \mathcal{A}} \left\{ R(s, a) + \gamma \mathbb{E}_{s'|s,a} \left[ V^*(s') \right] \right\}, \tag{2}$$

which can be formulated as a linear program (Puterman, 2014; Bertsekas, 1995):

$$P^* := \min_V \quad (1 - \gamma) \mathbb{E}_{s \sim \mu(s)} \left[ V(s) \right] \tag{3}$$

$$\text{s.t.} \quad V(s) \geqslant R(s, a) + \gamma \mathbb{E}_{s'|s,a} \left[ V(s') \right], \quad \forall (s, a) \in \mathcal{S} \times \mathcal{A}.$$

For completeness, we provide the derivation of the above equivalence in Appendix A. Without loss of generality, we assume there exists an optimal policy for the given MDP, namely, the linear programming is solvable. The optimal policy can be obtained from the solution to the linear program (3) via

$$\pi^*(s) = \underset{a \in \mathcal{A}}{\operatorname{argmax}} \left\{ R(s, a) + \gamma \mathbb{E}_{s'|s,a} \left[ V^*(s') \right] \right\}. \tag{4}$$

The dual form of the LP below is often easier to solve and yield more direct relations to the optimal policy.

$$D^* := \max_{\rho \geqslant 0} \quad \sum_{(s,a) \in \mathcal{S} \times \mathcal{A}} R(s, a) \rho(s, a) \tag{5}$$

$$\text{s.t.} \quad \sum_{a \in \mathcal{A}} \rho(s', a) = (1 - \gamma) \mu(s') + \gamma \sum_{s,a \in \mathcal{S} \times \mathcal{A}} \rho(s, a) P(s'|s, a) ds, \forall s' \in \mathcal{S}.$$

Since the primal LP is solvable, the dual LP is also solvable, and $P^* - D^* = 0$. The optimal dual variables $\rho^*(s, a)$ and optimal policy $\pi^*(a|s)$ are closely related in the following manner:

**Theorem 1 (Policy from dual variables)** $\sum_{s,a\in\mathcal{S}\times\mathcal{A}}\rho^*(s,a)=1$, and $\pi^*(a|s)=\frac{\rho^*(s,a)}{\sum_{a\in\mathcal{A}}\rho^*(s,a)}$.

Since the goal of reinforcement learning is to learn an optimal policy, it is appealing to deal with the Lagrangian dual which optimizes the policy directly, or its equivalent saddle point problem that jointly learns the optimal policy and value function.

**Theorem 2 (Competition in one-step setting)** *The optimal policy $\pi^*$, actor, and its corresponding value function $V^*$, dual critic, is the solution to the following saddle-point problem*

$$\max_{\alpha\in\mathcal{P}(\mathcal{S}),\pi\in\mathcal{P}(\mathcal{A})}\min_{V}\ L(V,\alpha,\pi):=(1-\gamma)\,\mathbb{E}_{s\sim\mu(s)}\left[V(s)\right]+\sum_{(s,a)\in\mathcal{S}\times\mathcal{A}}\alpha(s)\pi\left(a|s\right)\Delta[V](s,a),\ (6)$$

*where $\Delta[V](s,a):=R(s,a)+\gamma\mathbb{E}_{s'|s,a}[V(s')]-V(s)$.*

The saddle point optimization (6) provides a game perspective in understanding the reinforcement learning problem (Goodfellow et al., 2014). The learning procedure can be thought as a game between the dual critic, *i.e.*, value function for optimal policy, and the weighted actor, *i.e.*, $\alpha(s)\pi(a|s)$: the dual critic $V$ seeks the value function to satisfy the Bellman equation, while the actor $\pi$ tries to generate state-action pairs that break the satisfaction. Such a competition introduces new roles for the actor and the dual critic, and more importantly, bypasses the unnecessary separation of policy evaluation and policy improvement procedures needed in a traditional actor-critic framework.

## 3   SOURCES OF INSTABILITY

To solve the dual problem in (6), a straightforward idea is to apply stochastic mirror prox (Nemirovski et al., 2009) or stochastic primal-dual algorithm (Chen et al., 2014) to address the saddle point problem in (6). Unfortunately, such algorithms have limited use beyond special cases. For example, for an MDP with finite state and action spaces, the one-step saddle-point problem (6) with tabular parametrization is convex-concave, and finite-sample convergence rates can be established; see e.g., Chen & Wang (2016) and Wang (2017). However, when the state/action spaces are large or continuous so that function approximation must be used, such convergence guarantees no longer hold due to lack of convex-concavity. Consequently, directly solving (6) can suffer from severe bias and numerical issues, resulting in poor performance in practice (see, *e.g.*, Figure 1):

1. **Large bias in one-step Bellman operator**: It is well-known that one-step bootstrapping in temporal difference algorithms has lower variance than Monte Carlo methods and often require much fewer samples to learn. But it produces biased estimates, especially when function approximation is used. Such a bias is especially troublesome in our case as it introduces substantial noise in the gradients to update the policy parameters.

2. **Absence of local convexity and duality**: Using nonlinear parametrization will easily break the local convexity and duality between the original LP and the saddle point problem, which are known as the necessary conditions for the success of applying primal-dual algorithm to constrained problems (Luenberger & Ye, 2015). Thus none of the existing primal-dual type algorithms will remain stable and convergent when directly optimizing the saddle point problem without local convexity.

3. **Biased stochastic gradient estimator with under-fitted value function**: In the absence of local convexity, the stochastic gradient w.r.t. the policy $\pi$ constructed from under-fitted value function will presumably be biased and futile to provide any meaningful improvement of the policy. Hence, naively extending the stochastic primal-dual algorithms in Chen & Wang (2016); Wang (2017) for the parametrized Lagrangian dual, will also lead to biased estimators and sample inefficiency.

## 4   DUAL ACTOR-CRITIC

In this section, we will introduce several techniques to bypass the three instability issues in the previous section: (1) generalization of the minimax game to the multi-step case to achieve a better bias-variance tradeoff; (2) use of *path regularization* in the objective function to promote local convexity and duality; and (3) use of stochastic *dual ascent* to ensure unbiased gradient estimates.

### 4.1 COMPETITION IN MULTI-STEP SETTING

In this subsection, we will extend the minimax game between the actor and critic to the multi-step setting, which has been widely utilized in temporal-difference algorithms for better bias/variance tradeoffs (Sutton & Barto, 1998; Kearns & Singh, 2000). By the definition of the optimal value function, it is easy to derive the $k$-step Bellman optimality equation as

$$V^*(s) = \left(\mathcal{T}^k V^*\right)(s) := \max_{\pi \in \mathcal{P}} \left\{ \mathbb{E}^\pi \left[ \sum_{i=0}^k \gamma^i R(s_i, a_i) \right] + \gamma^{k+1} \mathbb{E}^\pi \left[ V^*(s_{k+1}) \right] \right\}. \quad (7)$$

Similar to the one-step case, we can reformulate the multi-step Bellman optimality equation into a form similar to the LP formulation, and then we establish the duality, which leads to the following mimimax problem:

**Theorem 3 (Competition in multi-step setting)** *The optimal policy $\pi^*$ and its corresponding value function $V^*$ is the solution to the following saddle point problem*

$$\max_{\alpha \in \mathcal{P}(\mathcal{S}), \pi \in \mathcal{P}(\mathcal{A})} \min_V L_k(V, \alpha, \pi) = (1 - \gamma^{k+1}) \mathbb{E}_\mu \left[ V(s) \right] + \mathbb{E}_\alpha^\pi \left[ \delta \left( \{s_i, a_i\}_{i=0}^k, s_{k+1} \right) \right], \quad (8)$$

*where $\delta \left( \{s_i, a_i\}_{i=0}^k, s_{k+1} \right) = \sum_{i=0}^k \gamma^i R(s_i, a_i) + \gamma^{k+1} V(s_{k+1}) - V(s)$ and*

$$\mathbb{E}_\alpha^\pi \left[ \delta \left( \{s_i, a_i\}_{i=0}^k, s_{k+1} \right) \right] = \sum_{\{s_i, a_i\}_{i=0}^k, s_{k+1}} \alpha(s_0) \prod_{i=0}^k \pi(a_i|s_i) p(s_{i+1}|s_i, a_i) \delta \left( \{s_i, a_i\}_{i=0}^k, s_{k+1} \right).$$

The saddle-point problem (8) is similar to the one-step Lagrangian (6): the dual critic, $V$, and weighted $k$-step actor, $\alpha(s_0) \prod_{i=0}^k \pi(a_i|s_i)$, are competing for an equilibrium, in which critic and actor become the optimal value function and optimal policy. However, it should be emphasized that due to the existence of max-operator over the space of distributions $\mathcal{P}(\mathcal{A})$, rather than $\mathcal{A}$, in the multi-step Bellman optimality equation (7), the establishment of the competition in multi-step setting in Theorem 3 is not straightforward: **i)**, its corresponding optimization is no longer a linear programming; **ii)**, the strong duality in (8) is not obvious because of the lack of the convex-concave structure. We first generalize the duality to multi-step setting. Due to space limit, detailed analyses for generalizing the competition to multi-step setting are provided in Appendix B.

### 4.2 PATH REGULARIZATION

When function approximation is used, the one-step or multi-step saddle-point problems (8) will no longer be convex in the primal parameters. This could lead to instability and even divergence when solved by brute-force stochastic primal-dual algorithms. One then desires to partially convexify the objectives without affecting the optimal solutions. The augmented Lagrangian method (Boyd et al., 2010; Luenberger & Ye, 2015), also known as *the method of multipliers*, is designed and widely used for such purposes. However, directly applying this method would require introducing penalty functions of the multi-step Bellman operator, which renders extra complexity and challenges in optimization. Interested readers are referred to Appendix B.2 for further details.

Instead, we propose to use *path regularization*, as a stepping stone for promoting local convexity and computation efficiency. The regularization term is motivated by the fact that the optimal value function satisfies the constraint $V(s) = \mathbb{E}^{\pi^*} \left[ \sum_{i=0}^\infty \gamma^i R(s_i, a_i) | s \right]$. In the same spirit as augmented Lagrangian, we will introduce to the objective the simple penalty function $\mathbb{E}_{s \sim \mu(s)} \left[ \left( \mathbb{E}^{\pi_b} \left[ \sum_{i=0}^\infty \gamma^i R(s_i, a_i) \right] - V(s) \right)^2 \right]$, leading to the following:

$$\begin{aligned} L_r(V, \alpha, \pi) \quad := \quad & (1 - \gamma^{k+1}) \mathbb{E}_\mu \left[ V(s) \right] + \mathbb{E}_\alpha^\pi \left[ \delta \left( \{s_i, a_i\}_{i=0}^k, s_{k+1} \right) \right] \\ & + \eta_V \mathbb{E}_{s \sim \mu(s)} \left[ \left( \mathbb{E}^{\pi_b} \left[ \sum_{i=0}^\infty \gamma^i R(s_i, a_i) \right] - V(s) \right)^2 \right], \end{aligned} \quad (9)$$

where $\eta_V \geqslant 0$ is a hyper-parameter controlling the strength of the regularization.

Note that in the penalty function above we use a behavior policy $\pi_b$ instead of an optimal policy, since the latter is unknown. Adding such a regularization enables local duality in the primal parameters. Indeed, this can be easily verified by showing the positive definiteness of the Hessian at a local solution. We call this approach *path regularization*, since it exploits the rewards in the *sample path*

to regularize the *solution path* of value function $V$ in the optimization procedure. As a by-product, the regularization also provides a mechanism to utilize *off-policy* samples from behavior policy $\pi_b$.

One can also see that the regularization indeed provides guidance and preference to search for the solution path. Specifically, in each update of $V$ the learning procedure, it tries to move towards the optimal value function while staying close to the value function of the behavior policy $\pi_b$. Intuitively, such regularization restricts the feasible domain of candidates $V$ to be a ball centered at $V^{\pi_b}$. Besides enhancing local convexity, such a penalty also avoids unboundedness of $V$ in the learning procedure, and thus more numerical robust. As long as the optimal value function is indeed in such a region, the introduced side-effect can be controlled. Formally, we can show that with appropriate $\eta_V$, the optimal solution $(V^*, \alpha^*, \pi^*)$ is not affected. The main results of this subsection are summarized by the following theorem.

**Theorem 4 (Property of path regularization)** *The local duality holds for $L_r(V, \alpha, \pi)$. Denote $(V^*, \alpha^*, \pi^*)$ as the solution to Bellman optimality equation, with some appropriate $\eta_V$, $(V^*, \alpha^*, \pi^*) = \mathrm{argmax}_{\alpha \in \mathcal{P}(\mathcal{S}), \pi \in \mathcal{P}(\mathcal{A})} \, \mathrm{argmin}_V \, L_r(V, \alpha, \pi).$*

The proof of the theorem is given in Appendix B.3. We emphasize that the theorem holds when $V$ is given enough capacity, *i.e.*, in the nonparametric limit. With parametrization introduced, definitely approximation error will be introduced, and the valid range of $\eta_V$, which keeps optimal solution unchanged, will be affected. However, the function approximation error is still an open problem for general class of parametrization, we omit such discussion here which is out of the range of this paper.

### 4.3    STOCHASTIC DUAL ASCENT UPDATE

Rather than the primal form, *i.e.*, $\min_V \max_{\alpha \in \mathcal{P}(\mathcal{S}), \pi \in \mathcal{P}(\mathcal{A})} L_r(V, \alpha, \pi)$, we focus on optimizing the dual form $\max_{\alpha \in \mathcal{P}(\mathcal{S}), \pi \in \mathcal{P}(\mathcal{A})} \min_V L_r(V, \alpha, \pi)$. The major reason is due to the sample efficiency consideration. In the primal form, to apply the stochastic gradient descent algorithm at $V^t$, one needs to solve $\max_{\alpha \in \mathcal{P}(\mathcal{S}), \pi \in \mathcal{P}(\mathcal{A})} L_r(V^t, \alpha, \pi)$ which involves sampling from each $\pi$ and $\alpha$ during the solution path for the subproblem. We define the regularized dual function $\ell_r(\alpha, \pi) := \min_V L_r(V, \alpha, \pi)$. We first show the unbiased gradient estimator of $\ell_r$ w.r.t. $\theta_\rho = (\theta_\alpha, \theta_\pi)$, which are parameters associated with $\alpha$ and $\pi$. Then, we incorporate the stochastic update rule to the dual ascent algorithm (Boyd et al., 2010), resulting in the *dual actor-critic* (Dual-AC) algorithm.

The gradient estimators of the dual functions can be derived using chain rule and are provided below.

**Theorem 5** *The regularized dual function $\ell_r(\alpha, \pi)$ has gradients estimators*

$$\nabla_{\theta_\alpha} \ell_r(\theta_\alpha, \theta_\pi) = \mathbb{E}_\alpha^\pi \left[ \delta\left(\{s_i, a_i\}_{i=0}^k, s_{k+1}\right) \nabla_{\theta_\alpha} \log \alpha(s) \right], \tag{10}$$

$$\nabla_{\theta_\pi} \ell_r(\theta_\alpha, \theta_\pi) = \mathbb{E}_\alpha^\pi \left[ \delta\left(\{s_i, a_i\}_{i=0}^k, s_{k+1}\right) \sum_{i=0}^k \nabla_{\theta_\pi} \log \pi(a|s) \right]. \tag{11}$$

Therefore, we can apply stochastic mirror descent algorithm with the gradient estimator given in Theorem 5 to the regularized dual function $\ell_r(\alpha, \pi)$. Since the dual variables are probabilistic distributions, it is natural to use $KL$-divergence as the prox-mapping to characterize the geometry in the family of parameters (Amari & Nagaoka, 1993; Nemirovski et al., 2009). Specifically, in the $t$-th iteration,

$$\theta_\rho^t = \mathrm{argmin}_{\theta_\rho} -\theta_\rho^\top \hat{g}_\rho^{t-1} + \frac{1}{\zeta_t} KL(\rho_{\theta_\rho}(s, a) \| \rho_{\theta_{\rho^{t-1}}}(s, a)), \tag{12}$$

where $\hat{g}_\rho^{t-1} = \widehat{\nabla}_{\theta_\rho} \ell_r\left(\theta_\alpha^{t-1}, \theta_\pi^{t-1}\right)$ denotes the stochastic gradients estimated through (10) and (11) via given samples and $KL(q(s, a) \| p(s, a)) = \int q(s, a) \log \frac{q(s,a)}{p(s,a)} ds da$. Intuitively, such update rule emphasizes a trade-off between the current policy and possible improvements based on samples. The update of $\pi$ shares some similarity to the TRPO, which is derived to ensure monotonic improvement of the new policy Schulman et al. (2015a). We discuss the details in Section 4.4.

Rather than just update $V$ once via the stochastic gradient of $\nabla_V L_r(V, \alpha, \pi)$ in each iteration for solving saddle-point problem (Nemirovski et al., 2009), which is only valid in convex-concave setting, Dual-AC exploits the stochastic dual ascent algorithm which requires $V^t =$

---

**Algorithm 1** Dual Actor-Critic (Dual-AC)

---

1: Initialize $\theta_V^0$, $\theta_\alpha^0$ and $\theta_\pi^0$ randomly, set $\beta \in [\frac{1}{2}, 1]$.
2: **for** episode $t = 1, \ldots, T$ **do**
3:     Start from $s \sim \alpha^{t-1}(s)$, collect samples $\{\tau_l\}_{l=1}^m$ follows behavior policy $\pi^{t-1}$.
4:     Update $\theta_V^t = \mathrm{argmin}_{\theta_V} \hat{L}_r(V, \alpha^{t-1}, \pi^{t-1})$ by SGD based on $\{\tau_l\}_{l=1}^m$.
5:     Update $\tilde{\alpha}^t(s)$ according to closed-form (14).
6:     Decay the stepsize: $\zeta_t = \frac{C}{n_0 + 1/t^\beta}$.
7:     Compute the stochastic gradients for $\theta_\pi$ following (11).
8:     Update $\theta_\pi^t$ according to the exact prox-mapping (16) or the approximate closed-form (17).
9: **end for**

---

$\mathrm{argmin}_V L_r(V, \alpha^{t-1}, \pi^{t-1})$ in $t$-th iteration for estimating $\nabla_{\theta_\rho} \ell_r(\theta_\alpha, \theta_\pi)$. As we discussed, such operation will keep the gradient estimator of dual variables unbiased, which provides better direction for convergence.

In Algorithm 1, we update $V^t$ by solving optimization $\min_V L_r(V, \alpha^{t-1}, \pi^{t-1})$. In fact, the $V$ function in the path-regularized Lagrangian $L_r(V, \alpha, \pi)$ plays two roles: **i)**, inherited from the original Lagrangian, the first two terms in regularized Lagrangian (9) push the $V$ towards the value function of the optimal policy with *on-policy* samples; **ii)**, on the other hand, the path regularization enforces $V$ to be close to the value function of behavior policy $\pi_b$ with *off-policy* samples. Therefore, the $V$ function in the Dual-AC algorithm can be understood as an interpolation between these two value functions learned from both on and off policy samples.

## 4.4 PRACTICAL IMPLEMENTATION

In above, we have introduced path regularization for recovering local duality property of the parametrized multi-step Lagrangian dual form and tailored stochastic mirror descent algorithm for optimizing the regularized dual function. Here, we present several strategies for practical computation considerations.

**Update rule of $V^t$.** In each iteration, we need to solve $V^t = \mathrm{argmin}_{\theta_V} L_r(V, \alpha^{t-1}, \pi^{t-1})$, which depends on $\pi_b$ and $\eta_V$, for estimating the gradient for dual variables. In fact, the closer $\pi_b$ to $\pi^*$ is, the smaller $\mathbb{E}_{s \sim \mu(s)} \left[ \left( \mathbb{E}^{\pi_b} \left[ \sum_{i=0}^\infty \gamma^i R(s_i, a_i) \right] - V^*(s) \right)^2 \right]$ will be. Therefore, we can set $\eta_V$ to be large for better local convexity and faster convergence. Intuitively, the $\pi^{t-1}$ is approaching to $\pi^*$ as the algorithm iterates. Therefore, we can exploit the policy obtained in previous iteration, *i.e.*, $\pi^{t-1}$, as the behavior policy. The experience replay can also be used.

Furthermore, notice the $L(V, \alpha^{t-1}, \pi^{t-1})$ is a expectation of functions of $V$, we will use stochastic gradient descent algorithm for the subproblem. Other efficient optimization algorithms can be used too. Specifically, the unbiased gradient estimator for $\nabla_{\theta_V} L(V, \alpha^{t-1}, \pi^{t-1})$ is

$$\nabla_{\theta_V} L_r(V, \alpha^{t-1}, \pi^{t-1}) = (1 - \gamma^{k+1}) \mathbb{E}_\mu \left[ \nabla_{\theta_V} V(s) \right] + \mathbb{E}_\alpha^\pi \left[ \nabla_{\theta_V} \delta \left( \{s_i, a_i\}_{i=0}^k, s_{k+1} \right) \right] \quad (13)$$
$$- 2\eta_V \mathbb{E}_\mu^{\pi_b} \left[ \left( \sum_{i=0}^\infty \gamma^i R(s_i, a_i) - V(s) \right) \nabla_{\theta_V} V(s) \right].$$

We can use $k$-step Monte Carlo approximation for $\mathbb{E}_\mu^{\pi_b} \left[ \sum_{i=0}^\infty \gamma^i R(s_i, a_i) \right]$ in the gradient estimator. As $k$ is large enough, the truncate error is negligible (Sutton & Barto, 1998). We will iterate via $\theta_V^{t,i} = \theta_V^{t,i-1} + \kappa_i \widehat{\nabla}_{\theta_V^{t,i-1}} L_r(V, \alpha^{t-1}, \pi^{t-1})$ until the algorithm converges.

It should be emphasized that in our algorithm, $V^t$ is not trying to approximate the value or advantage function of $\pi^t$, in contrast to most actor-critic algorithms. Although $V^t$ eventually becomes an approximation of the optimal value function once the solution reaches the global optimum, in each update $V^t$ is merely a function that helps the current policy to be improved. From this perspective, the Dual-AC bypasses the policy evaluation step.

**Update rule of $\alpha^t$.** In practice, we may face with the situation that the initial sampling distribution is fixed, *e.g.*, in MuJoCo tasks. Therefore, we cannot obtain samples from $\alpha^t(s)$ at each iteration. We assume that $\exists \eta_\mu \in (0, 1]$, such that $\alpha(s) = (1 - \eta_\mu)\beta(s) + \eta_\mu \mu(s)$ with $\beta(s) \in \mathcal{P}(\mathcal{S})$. Hence, we have

$$\mathbb{E}_\alpha^\pi \left[ \delta \left( \{s_i, a_i\}_{i=0}^k, s_{k+1} \right) \right] = \mathbb{E}_\mu^\pi \left[ (\tilde{\alpha}(s) + \eta_\mu) \delta \left( \{s_i, a_i\}_{i=0}^k, s_{k+1} \right) \right],$$

where $\tilde{\alpha}(s) = (1 - \eta_\mu)\frac{\beta(s)}{\mu(s)}$. Note that such an assumption is much weaker comparing with the requirement for popular policy gradient algorithms (*e.g.*, Sutton et al. (1999); Silver et al. (2014)) that assumes $\mu(s)$ to be a stationary distribution. In fact, we can obtain a closed-form update for $\tilde{\alpha}$ if a square-norm regularization term is introduced into the dual function. Specifically,

**Theorem 6** *In $t$-th iteration, given $V^t$ and $\pi^{t-1}$,*

$$\operatorname*{argmax}_{\alpha \geqslant 0} \mathbb{E}_{\mu(s)\pi^{t-1}(s)} \left[ (\tilde{\alpha}(s) + \eta_\mu)\, \delta\left( \{s_i, a_i\}_{i=0}^k, s_{k+1} \right) \right] - \eta_\alpha \|\tilde{\alpha}\|_\mu^2 \tag{14}$$

$$= \frac{1}{\eta_\alpha} \max\left( 0, \mathbb{E}^{\pi^{t-1}} \left[ \delta\left( \{s_i, a_i\}_{i=0}^k, s_{k+1} \right) \right] \right). \tag{15}$$

Then, we can update $\tilde{\alpha}^t$ through (14) with Monte Carlo approximation of $\mathbb{E}^{\pi^{t-1}} \left[ \delta\left( \{s_i, a_i\}_{i=0}^k, s_{k+1} \right) \right]$, avoiding the parametrization of $\tilde{\alpha}$. As we can see, the $\tilde{\alpha}^t(s)$ reweights the samples based on the temporal differences and this offers a principled justification for the heuristic prioritized reweighting trick used in (Schaul et al., 2015).

**Update rule of $\theta_\pi^t$.** The parameters for dual function, $\theta_\rho$, are updated by the prox-mapping operator (12) following the stochastic mirror descent algorithm for the regularized dual function. Specifically, in $t$-th iteration, given $V^t$ and $\alpha^t$, for $\theta_\pi$, the prox-mapping (12) reduces to

$$\theta_\pi^t = \operatorname{argmin}_{\theta_\pi} -\theta_\pi^\top \hat{g}_\pi^t + \frac{1}{\zeta_t} KL\left( \pi_{\theta_\pi}(a|s) || \pi_{\theta_\pi^{t-1}}(a|s) \right), \tag{16}$$

where $\hat{g}_\pi^t = \widehat{\nabla}_{\theta_\pi} \ell_r\left( \theta_\alpha^t, \theta_\pi^t \right)$. Then, the update rule will become exactly the natural policy gradient (Kakade, 2002) with a principled way to compute the "policy gradient" $\hat{g}_\pi^t$. This can be understood as the penalty version of the trust region policy optimization (Schulman et al., 2015a), in which the policy parameters conservative update in terms of $KL$-divergence is achieved by adding explicit constraints.

Exactly solving the prox-mapping for $\theta_\pi$ requires another optimization, which may be expensive. To further accelerate the prox-mapping, we approximate the KL-divergence with the second-order Taylor expansion, and obtain an approximate closed-form update given by

$$\theta_\pi^t \approx \operatorname*{argmin}_{\theta_\pi} \left\{ -\theta_\pi^\top \hat{g}_\pi^t + \frac{1}{2} \left\| \theta_\pi - \theta_\pi^{t-1} \right\|_{F_t}^2 \right\} = \theta_\pi^{t-1} + \zeta_t F_t^{-1} \hat{g}_\pi^t, \tag{17}$$

where $F_t := \mathbb{E}_{\alpha^t \pi^{t-1}} \left[ \nabla^2 \log \pi_{\theta_\pi^{t-1}} \right]$ denotes the Fisher information matrix. Empirically, we may normalize the gradient by its norm $\sqrt{g_\pi^t F_t^{-1} g_\pi^t}$ (Rajeswaran et al., 2017) for better performances.

Combining these practical tricks to the stochastic mirror descent update eventually gives rise to the dual actor-critic algorithm outlined in Algorithm 1.

## 5 RELATED WORK

The dual actor-critic algorithm includes both the learning of *optimal* value function and *optimal* policy in a *unified* framework based on the duality of the linear programming (LP) representation of Bellman optimality equation. The linear programming formulation of Bellman optimality equation and its duality have been used for (approximate) planning problem (Schweitzer & Seidmann, 1985; de Farias & Van Roy, 2003; Wang et al., 2007; O'Donoghue et al., 2011; Cogill, 2015), in which the transition probability of the MDP is known and the value function or policy are in tabular form. Chen & Wang (2016); Wang (2017) apply stochastic first-order algorithms (Nemirovski et al., 2009) for the one-step Lagrangian of the LP problem in reinforcement learning setting. However, as we discussed in Section 3, their algorithm is restricted to tabular parametrization and are not applicable to MDPs with large or continuous state/action spaces. Other authors have also considered approximate LP formulations to solve large-scale problems Pazis & Parr (2011); Abbasi-Yadkori et al. (2014); Lakshminarayanan et al. (2017), but they either do not focus on concrete algorithms for solving the optimization problem, or require certain knowledge of the transition probability function that may be hard to obtain in practice.

The duality view has also been exploited in Neu et al. (2017). Their algorithm is based on the duality of entropy-regularized Bellman equation (Todorov, 2007; Rubin et al., 2012; Fox et al.,

2016; Haarnoja et al., 2017; Asadi & Littman, 2017; Nachum et al., 2017), rather than the exact Bellman optimality equation we try to solve in this work.

Our dual actor-critic algorithm can be understood as a nontrivial extension of the (approximate) dual gradient method (Bertsekas, 1999, Chapter 6.3) using stochastic gradient and Bregman divergence, which essentially parallels the view of (approximate) stochastic mirror descent algorithm (Nemirovski et al., 2009) in the primal space. As a result, the algorithm converges with diminishing stepsizes and decaying errors from solving subproblems.

Particularly, the update rules of $\alpha$ and $\pi$ in the dual actor-critic are related to several existing algorithms. As we see in the update of $\alpha$, the algorithm reweighs the samples which are not fitted well. This is related to the heuristic prioritized experience replay (Schaul et al., 2015). For the update in $\pi$, the proposed algorithm bears some similarities with trust region poicy gradient (TRPO) (Schulman et al., 2015a) and natural policy gradient (Kakade, 2002; Rajeswaran et al., 2017). Indeed, TRPO and NPR solve the same prox-mapping but are derived from different perspectives. We emphasize that although the updating rules share some resemblance to several reinforcement learning algorithms in the literature, they are purely originated from a stochastic dual ascent algorithm for solving the two-play game derived from Bellman optimality equation.

## 6 EXPERIMENTS

We evaluated the dual actor-critic (Dual-AC) algorithm on several continuous control environments from the OpenAI Gym (Brockman et al., 2016) with MuJoCo physics simulator (Todorov et al., 2012). We compared Dual-AC with several representative actor-critic algorithms, including trust region policy optimization (TRPO) (Schulman et al., 2015a) and proximal policy optimization (PPO) (Schulman et al., 2017)[1]. We ran the algorithms with 5 random seeds and reported the average rewards with 50% confidence interval. Details of the tasks and setups of these experiments including the policy/value function architectures and the hyperparameters values, are provided in Appendix C.

### 6.1 ABLATION STUDY

To justify our analysis in identifying the sources of instability in directly optimizing the parametrized one-step Lagrangian duality and the effect of the corresponding components in the dual actor-critic algorithm, we perform comprehensive Ablation study in InvertedDoublePendulum-v1, Swimmer-v1, and Hopper-v1 environments. We also considered the effect of $k = \{10, 50\}$ besides the one-step result in the study to demonstrate the benefits of multi-step.

We conducted comparison between the Dual-AC and its variants, including Dual-AC w/o multi-step, Dual-AC w/o path-regularization, Dual-AC w/o unbiased $V$, and the naive Dual-AC, for demonstrating the three instability sources in Section 3, respectively, as well as varying the $k = \{10, 50\}$ in Dual-AC. Specifically, Dual-AC w/o path-regularization removes the path-regularization components; Dual-AC w/o multi-step removes the multi-step extension and the path-regularization; Dual-AC w/o unbiased $V$ calculates the stochastic gradient without achieving the convergence of inner optimization on $V$; and the naive Dual-AC is the one without all components. Moreover, Dual-AC with $k = 10$ and Dual-AC with $k = 50$ denote the length of steps set to be 10 and 50, respectively.

The empirical performances on InvertedDoublePendulum-v1, Swimmer-v1, and Hopper-v1 tasks are shown in Figure 1. The results are consistent across the tasks with the analysis. The naive Dual-AC performs the worst. The performances of the Dual-AC found the optimal policy which solves the problem much faster than the alternative variants. The Dual-AC w/o unbiased $V$ converges slower, showing its sample inefficiency caused by the bias in gradient calculation. The Dual-AC w/o multi-step and Dual-AC w/o path-regularization cannot converge to the optimal policy, indicating the importance of the path-regularization in recovering the local duality. Meanwhile, the performance of Dual-AC w/o multi-step is worse than Dual-AC w/o path-regularization, showing the bias in one-

---

[1]As discussed in Henderson et al. (2018), different implementations of TRPO and PPO can provide different performances. For a fair comparison, we use the codes from https://github.com/joschu/modular_rl reported to have achieved the best scores in Henderson et al. (2018).

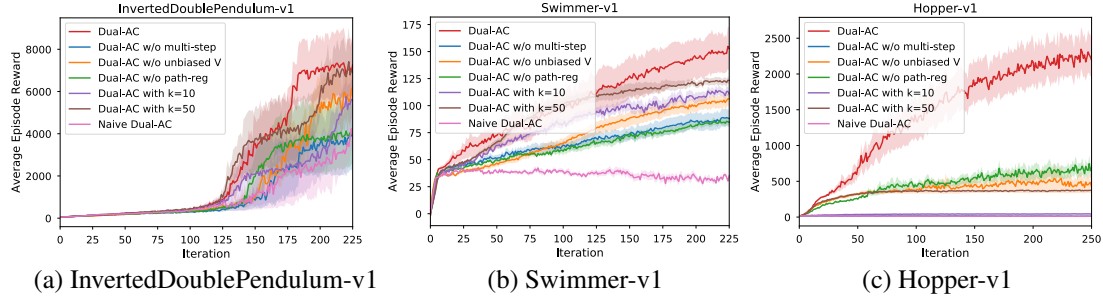

Figure 1: Comparison between the Dual-AC and its variants for justifying the analysis of the source of instability.

step can be alleviated via multi-step trajectories. The performances of Dual-AC become better with the length of step $k$ increasing on these three tasks. We conjecture that the main reason may be that in these three MuJoCo environments, the bias dominates the variance. Therefore, with the $k$ increasing, the proposed Dual-AC obtains more accumulate rewards.

## 6.2 COMPARISON IN CONTINUOUS CONTROL TASKS

In this section, we evaluated the Dual-AC against TRPO and PPO across multiple tasks, including the InvertedDoublePendulum-v1, Hopper-v1, HalfCheetah-v1, Swimmer-v1 and Walker-v1. These tasks have different dynamic properties, ranging from unstable to stable, Therefore, they provide sufficient benchmarks for testing the algorithms. In Figure 2, we reported the average rewards across 5 runs of each algorithm with 50% confidence interval during the training stage. We also reported the average final rewards in Table 1.

Table 1: The average final performances of the policies learned from Dual-AC and the competitors.

| Environment | Dual-AC | PPO | TRPO |
|---|---|---|---|
| Pendulum | $-155.45$ | $-266.98$ | $-245.11$ |
| InvertedDoublePendulum | $8599.47$ | $1776.26$ | $3070.96$ |
| Swimmer | $234.56$ | $223.13$ | $232.89$ |
| Hopper | $2983.79$ | $2376.15$ | $2483.57$ |
| HalfCheetah | $3041.47$ | $2249.10$ | $2347.19$ |
| Walker | $4103.60$ | $3315.45$ | $2838.99$ |

The proposed Dual-AC achieves the best performance in almost all environments, including Pendulum, InvertedDoublePendulum, Hopper, HalfCheetah and Walker. These results demonstrate that Dual-AC is a viable and competitive RL algorithm for a wide spectrum of RL tasks with different dynamic properties.

A notable case is the InvertedDoublePendulum, where Dual-AC substantially outperforms TRPO and PPO in terms of the learning speed and sample efficiency, implying that Dual-AC is preferable to unstable dynamics. We conjecture this advantage might come from the different meaning of $V$ in our algorithm. For unstable system, the failure will happen frequently, resulting the collected data are far away from the optimal trajectories. Therefore, the policy improvement through the value function corresponding to current policy is slower, while our algorithm learns the optimal value function and enhances the sample efficiency.

## 7 CONCLUSION

In this paper, we revisited the linear program formulation of the Bellman optimality equation, whose Lagrangian dual form yields a game-theoretic view for the roles of the actor and the dual critic. Although such a framework for actor and dual critic allows them to be optimized for the same objective function, parametering the actor and dual critic unfortunately induces instablity in optimization. We analyze the sources of instability, which is corroborated by numerical experiments. We then propose *Dual Actor-Critic* , which exploits *stochastic dual ascent* algorithm for the *path regularized*,

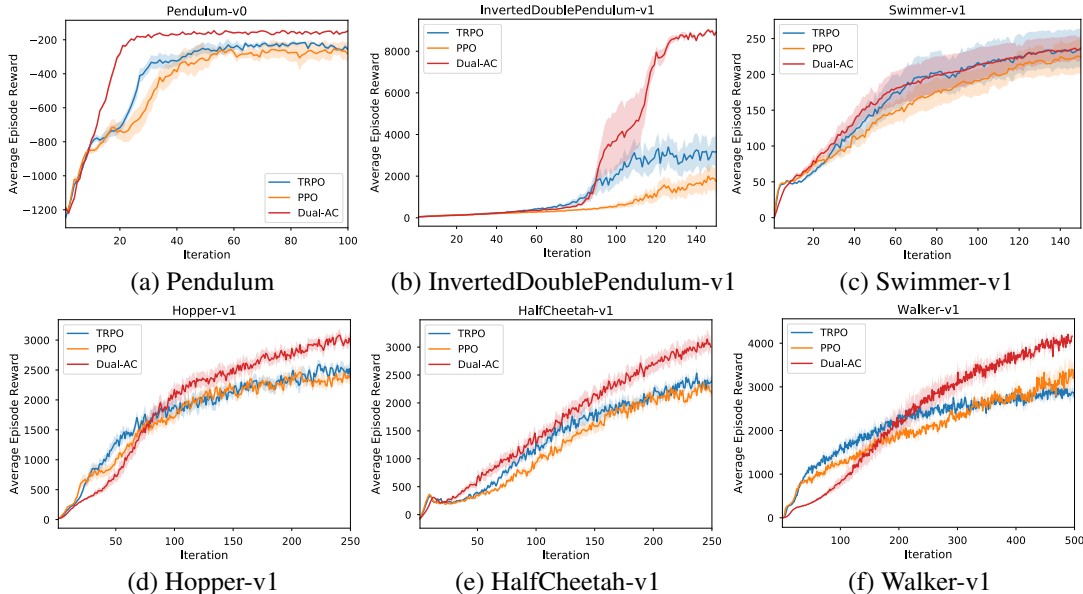

(a) Pendulum       (b) InvertedDoublePendulum-v1       (c) Swimmer-v1

(d) Hopper-v1       (e) HalfCheetah-v1       (f) Walker-v1

Figure 2: The results of Dual-AC against TRPO and PPO baselines. Each plot shows average reward during training across 5 random seeded runs, with 50% confidence interval. The x-axis is the number of training iterations. The Dual-AC achieves comparable performances comparing with TRPO and PPO in some tasks, but outperforms on more challenging tasks.

*multi-step bootstrapping* two-player game, to bypass these issues. The algorithm achieves the state-of-the-art performances on several MuJoCo benchmarks.

## ACKNOWLEDGMENTS

LS is supported in part by NSF IIS-1218749, NIH BIGDATA 1R01GM108341, NSF CAREER IIS-1350983, NSF IIS-1639792 EAGER, NSF CNS-1704701, ONR N00014-15-1-2340, Intel ISTC, NVIDIA and Amazon AWS. NH is supported by NSF CRII-1755829. We thank Mengdi Wang for helpful feedback and discussions.

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

# Appendix

## A DETAILS OF THE PROOFS FOR SECTION 2

### A.1 DUALITY OF BELLMAN OPTIMALITY EQUATION

Puterman (2014); Bertsekas (1995) provide details in deriving the linear programming form of the Bellman optimality equation. We provide a briefly proof here.

**Proof** We rewrite the linear programming 3 as

$$V^* = \operatorname*{argmin}_{V \geqslant \mathcal{T}V} \mathbb{E}_\mu [V(s)]. \tag{18}$$

Recall the $\mathcal{T}$ is monotonic, *i.e.*, if $V \geqslant \mathcal{T}V \Rightarrow \mathcal{T}V \geqslant \mathcal{T}^2 V$ and $V^* = \mathcal{T}^\infty V$ for arbitrary $V$, we have for $\forall V$ feasible, $V \geqslant \mathcal{T}V \geqslant \mathcal{T}^2 V \geqslant \ldots \geqslant \mathcal{T}^\infty V = V^*$. ∎

**Theorem 1 (Optimal policy from occupancy)** $\sum_{s,a \in \mathcal{S} \times \mathcal{A}} \rho^*(s,a) = 1$, *and* $\pi^*(a|s) = \frac{\rho^*(s,a)}{\sum_{a \in \mathcal{A}} \rho^*(s,a)}$.

**Proof** For the optimal occupancy measure, it must satisfy

$$\sum_{a \in \mathcal{A}} \rho^*(s',a) = \gamma \sum_{s,a \in \mathcal{S} \times \mathcal{A}} \rho^*(s,a) p(s'|s,a) + (1-\gamma)\mu(s'), \quad \forall s' \in \mathcal{S}$$

$$\Rightarrow \quad (1-\gamma)\mu + \sum_{s,a \in \mathcal{S} \times \mathcal{A}} (\gamma P - I)\rho^*(s,a) = 0,$$

where $P$ denotes the transition distribution and $I$ denotes a $|\mathcal{S}| \times |\mathcal{SA}|$ matrix where $I_{ij} = 1$ if and only if $j \in [(i-1)|\mathcal{A}| + 1, \ldots, i|\mathcal{A}|]$. Multiply both sides with $\mathbf{1}$, due to $\mu$ and $P$ are probabilities, we have $\langle \mathbf{1}, \rho^* \rangle = 1$.

Without loss of generality, we assume there is only one best action in each state. Therefore, by the KKT complementary conditions of (3), *i.e.*,

$$\rho(s,a)\left(R(s,a) + \gamma \mathbb{E}_{s'|s,a}[V(s')] - V(s)\right) = 0,$$

which implies $\rho^*(s,a) \neq 0$ if and only if $a = a^*$, therefore, the $\pi^*$ by normalization. ∎

**Theorem 2** *The optimal policy* $\pi^*$ *and its corresponding value function* $V^*$ *is the solution to the following saddle problem*

$$\max_{\alpha \in \mathcal{P}(\mathcal{S}), \pi \in \mathcal{P}(\mathcal{A})} \min_V L(V, \alpha, \pi) := (1-\gamma)\mathbb{E}_{s \sim \mu(s)}[V(s)] + \sum_{(s,a) \in \mathcal{S} \times \mathcal{A}} \alpha(s)\pi(a|s)\Delta[V](s,a)$$

*where* $\Delta[V](s,a) = R(s,a) + \gamma \mathbb{E}_{s'|s,a}[V(s')] - V(s)$.

**Proof** Due to the strong duality of the optimization (3), we have

$$\min_V \max_{\rho(s,a) \geqslant 0} (1-\gamma)\mathbb{E}_{s \sim \mu(s)}[V(s)] + \sum_{(s,a) \in \mathcal{S} \times \mathcal{A}} \rho(s,a)\Delta[V](s,a)$$

$$= \max_{\rho(s,a) \geqslant 0} \min_V (1-\gamma)\mathbb{E}_{s \sim \mu(s)}[V(s)] + \sum_{(s,a) \in \mathcal{S} \times \mathcal{A}} \rho(s,a)\Delta[V](s,a).$$

Then, plugging the property of the optimum in Theorem 1, we achieve the final optimization (6). ∎

### A.2 CONTINUOUS STATE AND ACTION MDP EXTENSION

In this section, we extend the linear programming and its duality to continuous state and action MDP. In general, the only weak duality holds for infinite constraints, *i.e.*, $P^* \geqslant D^*$. With a mild assumption, we will recover the strong duality for continuous state and action MDP, and most of the conclusions in discrete state and action MDP still holds.

Specifically, without loss of generality, we consider the solvable MDP, *i.e.*, the optimal policy, $\pi^*(a|s)$, exists. If $\|R(s,a)\|_\infty \leqslant C_R$, $\|V^*\|_\infty \leqslant \frac{C_R}{1-\gamma}$. Moreover,

$$
\begin{aligned}
\|V^*\|_{2,\mu}^2 &= \int (V^*(s))^2 \, \mu(s)ds = \int \left(R(s,a) + \gamma\mathbb{E}_{s'|s,a}\left[V^*(s')\right]\right)^2 \pi^*(a|s)\mu(s)d(s,a) \\
&\leqslant 2\int (R(s,a))^2 \, \pi^*(a|s)\mu(s)ds + 2\gamma^2 \int \left(\mathbb{E}_{s'|s,a}\left[V^*(s')\right]\right)^2 \pi^*(a|s)\mu(s)ds \\
&\leqslant 2\max_{a\in\mathcal{A}} \|R(s,a)\|_\mu^2 + 2\gamma^2 \int \left(\int P^*(s'|s)\mu(s)ds\right)(V^*(s'))^2 \, ds' \\
&\leqslant 2\max_{a\in\mathcal{A}} \|R(s,a)\|_\mu^2 + 2\gamma^2 \|V^*(s')\|_\infty^2 \int\int P^*(s'|s)\mu(s)dsds' \\
&\leqslant 2\max_{a\in\mathcal{A}} \|R(s,a)\|_\mu^2 + 2\gamma^2 \|V^*(s')\|_\infty^2 ,
\end{aligned}
$$

where the first inequality comes from $2\langle f(x), g(x)\rangle_2 \leqslant \|f\|_2^2 + \|g\|_2^2$.

$$
\left\|V^* - \gamma\mathbb{E}_{s'|s,a}\left[V(s')\right]\right\|_{\mu\pi_b}^2 \leqslant 2\|V^*\|_\mu^2 + 2\gamma^2 \left\|\mathbb{E}_{s'|s,a}\left[V(s')\right]\right\|_{\mu\pi_b}^2 \leqslant 2\|V^*\|_\mu^2 + 2\gamma^2 \|V^*(s')\|_\infty^2 ,
$$

for some $\pi_b \in \mathcal{P}$ that $\pi_b(a|s) > 0$ for $\forall (s,a) \in \mathcal{S} \times \mathcal{A}$. Therefore, with the assumption that $\|R(s,a)\|_\mu^2 \leqslant C_R^\mu, \forall a \in \mathcal{A}$, we have $R(s,a) \in \mathcal{L}_{\mu\pi_b}^2 (\mathcal{S} \times \mathcal{A})$ and $V^*(s') \in \mathcal{L}_\mu^2(\mathcal{S})$. The constraints in the primal form of linear programming can be written as

$$
(\mathcal{I} - \gamma\mathcal{P})\, V - R \succeq_{\mathcal{L}_{\mu\pi_b}^2} 0,
$$

where $\mathcal{I} - \gamma\mathcal{P} : \mathcal{L}_\mu^2(\mathcal{S}) \to \mathcal{L}_{\mu\pi_b}^2(\mathcal{S} \times \mathcal{A})$ without any effect on the optimality. For simplicity, we denote $\succeq$ as $\succeq_{\mathcal{L}_{\mu\pi_b}^2}$ and $\langle f, g\rangle = \int f(s,a)g(s,a)\mu(s)\pi_b(a|s)dsda$. Apply the Lagrangian multiplier for constraints in ordered Banach space in Burger (2003), we have

$$
\mathsf{P}^* = \min_{V\in\mathcal{L}} \max_{\varrho\succeq 0} (1-\gamma)\mathbb{E}_\mu\left[V(s)\right] - \langle\varrho, (\mathcal{I} - \gamma\mathcal{P})\, V - R\rangle. \tag{19}
$$

The solution $(V^*, \varrho^*)$ also satisfies the KKT conditions,

$$
\begin{aligned}
(1-\gamma)\mathbf{1} - (\mathcal{I} - \gamma\mathcal{P})^\top \varrho^* &= 0, \tag{20}\\
\varrho^* &\succeq 0, \tag{21}\\
(\mathcal{I} - \gamma\mathcal{P})\, V^* - R &\succeq 0, \tag{22}\\
\langle\varrho^*, (\mathcal{I} - \gamma\mathcal{P})\, V^* - R\rangle &= 0. \tag{23}
\end{aligned}
$$

where $^\top$ denotes the conjugate operation. By the KKT condition, we have

$$
\left\langle\mathbf{1}, (1-\gamma)\mathbf{1} - (\mathcal{I} - \gamma\mathcal{P})^\top \varrho^*\right\rangle = 0 \Rightarrow \langle\mathbf{1}, \varrho\rangle = 1. \tag{24}
$$

The strongly duality also holds, *i.e.*,

$$
\mathsf{P}^* = \mathsf{D}^* := \max_{\varrho\succeq 0} \quad \langle R(s,a), \varrho(s,a)\rangle \tag{25}
$$

$$
\text{s.t.} \quad (1-\gamma)\mathbf{1} - (\mathcal{I} - \gamma\mathcal{P})^\top \varrho = 0 \tag{26}
$$

**Proof** We compute the duality gap

$$
\begin{aligned}
&(1-\gamma)\langle\mathbf{1}, V^*\rangle - \langle R, \varrho^*\rangle \\
={} &\langle\varrho^*, (\mathcal{I} - \gamma\mathcal{P})\, V^*\rangle - \langle R, \varrho^*\rangle \\
={} &\langle\varrho^*, (\mathcal{I} - \gamma\mathcal{P})\, V^* - R\rangle = 0,
\end{aligned}
$$

which shows the strongly duality holds. ∎

## B    DETAILS OF THE PROOFS FOR SECTION 4

### B.1    COMPETITION IN MULTI-STEP SETTING

Once we establish the $k$-step Bellman optimality equation (7), it is easy to derive the $\lambda$-Bellman optimality equation, *i.e.*,

$$V^*(s) = \max_{\pi \in \mathcal{P}} (1 - \lambda) \sum_{k=0}^{\infty} \lambda^k \mathbb{E}^{\pi} \left[ \sum_{i=0}^{k} \gamma^i R(s_i, a_i) + \gamma^{k+1} V^*(s_{k+1}) \right] := (\mathcal{T}_{\lambda} V^*)(s). \qquad (27)$$

**Proof**  Denote the optimal policy as $\pi^*(a|s)$, we have

$$V^*(s) = \mathbb{E}^{\pi^*}_{\{s_t\}_{i=0}|s} \left[ \sum_{i=0}^{k} \gamma^i R(s_i, a_i) \right] + \gamma^{k+1} \mathbb{E}^{\pi^*}_{s_{k+1}|s} \left[ V^*(s_{k+1}) \right],$$

holds for arbitrary $\forall k \in \mathbb{N}$. Then, we conduct $k \sim \mathcal{G}eo(\lambda)$ and take expectation over the countable infinite many equation, resulting

$$\begin{aligned}
V^*(s) &= (1 - \lambda) \sum_{k=0}^{\infty} \lambda^k \mathbb{E}^{\pi^*} \left[ \sum_{i=0}^{k} \gamma^i R(s_i, a_i) + \gamma^{k+1} V^*(s_{k+1}) \right] \\
&= \max_{\pi \in \mathcal{P}} (1 - \lambda) \sum_{k=0}^{\infty} \lambda^k \mathbb{E}^{\pi} \left[ \sum_{i=0}^{k} \gamma^i R(s_i, a_i) + \gamma^{k+1} V^*(s_{k+1}) \right]
\end{aligned}$$

∎

Next, we investigate the equivalent optimization form of the $k$-step and $\lambda$-Bellman optimality equation, which requires the following monotonic property of $\mathcal{T}_k$ and $\mathcal{T}_{\lambda}$.

**Lemma 7**  *Both $\mathcal{T}_k$ and $\mathcal{T}_{\lambda}$ are monotonic.*

**Proof**  Assume $U$ and $V$ are the value functions corresponding to $\pi_1$ and $\pi_2$, and $U \geqslant V$, *i.e.*, $U(s) \geqslant V(s), \forall s \in \mathcal{S}$, apply the operator $\mathcal{T}_k$ on $U$ and $V$, we have

$$\begin{aligned}
\left( \mathcal{T}_k U \right)(s) &= \max_{\pi \in \mathcal{P}} \mathbb{E}^{\pi}_{\{s_i\}_{i=1}^k|s} \left[ \sum_{i=0}^{k} \gamma^i R(s_i, a_i) \right] + \gamma^{k+1} \mathbb{E}^{\pi}_{s_{k+1}|s} \left[ U(s_{k+1}) \right], \\
\left( \mathcal{T}_k V \right)(s) &= \max_{\pi \in \mathcal{P}} \mathbb{E}^{\pi}_{\{s_i\}_{i=1}^k|s} \left[ \sum_{i=0}^{k} \gamma^i R(s_i, a_i) \right] + \gamma^{k+1} \mathbb{E}^{\pi}_{s_{k+1}|s} \left[ V(s_{k+1}) \right].
\end{aligned}$$

Due to $U \geqslant V$, we have $\mathbb{E}^{\pi}_{s_{k+1}|s} \left[ U(s_{k+1}) \right] \geqslant \mathbb{E}^{\pi}_{s_{k+1}|s} \left[ V(s_{k+1}) \right], \forall \pi \in \mathcal{P}$, which leads to the first conclusion, $\mathcal{T}_k U \geqslant \mathcal{T}_k V$.

Since $\mathcal{T}_{\lambda} = (1 - \lambda) \sum_{k=1}^{\infty} \mathcal{T}_k = \mathbb{E}_{k \sim \mathcal{G}eo(\lambda)} [\mathcal{T}_k]$, therefore, $\mathcal{T}_{\lambda}$ is also monotonic.    ∎

With the monotonicity of $\mathcal{T}_k$ and $\mathcal{T}_{\lambda}$, we can rewrite the $V^*$ as the solution to an optimization,

**Theorem 8**  *The optimal value function $V^*$ is the solution to the optimization*

$$V^* = \operatorname*{argmin}_{V \geqslant \mathcal{T}_k V} \left( 1 - \gamma^{k+1} \right) \mathbb{E}_{s \sim \mu(s)} \left[ V(s) \right], \qquad (28)$$

*where $\mu(s)$ is an arbitrary distribution over $\mathcal{S}$.*

**Proof**  Recall the $\mathcal{T}_k$ is monotonic, *i.e.*, $V \geqslant \mathcal{T}_k V \Rightarrow \mathcal{T}_k V \geqslant \mathcal{T}_k^2 V$ and $V^* = \mathcal{T}_k^{\infty} V$ for arbitrary $V$, we have for $\forall V$, $V \geqslant \mathcal{T}_k V \geqslant \mathcal{T}_k^2 V \geqslant \dots \geqslant \mathcal{T}_k^{\infty} V = V^*$, where the last equality comes from the Banach fixed point theorem (Puterman, 2014). Similarly, we can also show that $\forall V$, $V \geqslant \mathcal{T}_{\lambda}^{\infty} V = V^*$. By combining these two inequalities, we achieve the optimization.    ∎

We rewrite the optimization as

$$\min_{V} \quad (1 - \gamma^{k+1})\mathbb{E}_{s\sim\mu(s)}\left[V(s)\right] \tag{29}$$

$$\text{s.t.} \quad V(s) \geqslant R(s,a) + \max_{\pi\in\mathcal{P}} \mathbb{E}^{\pi}_{\{s_i\}_{i=1}^{k+1}|s}\left[\sum_{i=1}^{k}\gamma^i R(s_i, a_i) + \gamma^{k+1}V(s_{k+1})\right],$$

$$(s,a) \in \mathcal{S} \times \mathcal{A},$$

We emphasize that this optimization is no longer linear programming since the existence of $\max$-operator over distribution space in the constraints. However, Theorem 1 still holds for the dual variables in (32).

**Proof** Denote the optimal policy as $\tilde{\pi}_V^* = \text{argmax}_{\pi\in\mathcal{P}} \mathbb{E}^{\pi}_{\{s_i\}_{i=1}^{k+1}|s}\left[\sum_{i=1}^{k}\gamma^i R(s_i, a_i) + \gamma^{k+1}V(s_{k+1})\right]$, the KKT condition of the optimization (29) can be written as

$$\left(1 - \gamma^{k+1}\right)\mu(s') + \gamma^{k+1}\sum_{\{s_i,a_i\}_{i=0}^{k}} p(s'|s_k, a_k)\prod_{i=0}^{k-1} p(s_{i+1}|s_i, a_i)\prod_{i=1}^{k}\tilde{\pi}_V^*(a_i|s_i)\rho^*(s_0, a_0)$$

$$= \sum_{a_0,\{s_i,a_i\}_{i=1}^{k}}\prod_{i=0}^{k} p(s_{i+1}|s_i, a_i)\rho^*(s', a)\prod_{i=1}^{k}\tilde{\pi}_V^*(a_i|s_i).$$

Denote $P_k^{\pi}(s_{k+1}|s,a) = \sum_{\{s_i,a_i\}_{i=1}^{k}} p(s_{k+1}|s_k, a_k)\prod_{i=0}^{k-1} p(s_{i+1}|s_i, a_i)\prod_{i=1}^{k}\pi(a_i|s_i)$, we simplify the condition, *i.e.*,

$$\left(1 - \gamma^{k+1}\right)\mu(s') + \gamma^{k+1}\sum_{s,a} P_k^{\tilde{\pi}_V^*}(s'|s,a)\rho^*(s,a) = \sum_{a}\rho^*(s', a).$$

Due to the $P_k^{\pi_V^*}(s'|s,a)$ is a conditional probability for $\forall V$, with similar argument in Theorem 1, we have $\sum_{s,a}\rho^*(s,a) = 1$.

By the KKT complementary condition, the primal and dual solutions, *i.e.*, $V^*$ and $\rho^*$, satisfy

$$\rho^*(s,a)\left(R(s,a) + \mathbb{E}^{\tilde{\pi}_{V^*}^*}_{\{s_i\}_{i=1}^{k+1}|s}\left[\sum_{i=1}^{k}\gamma^i R(s_i, a_i) + \gamma^{k+1}V^*(s_{k+1})\right] - V^*(s)\right) = 0. \tag{30}$$

Recall $V^*$ denotes the value function of the optimal policy, then, based on the definition, $\tilde{\pi}_{V^*}^* = \pi^*$ which denotes the optimal policy. Then, the condition (30) implies $\rho(s,a) \neq 0$ if and only if $a = a^*$, therefore, we can decompose $\rho^*(s,a) = \alpha^*(s)\pi^*(a|s)$.

∎

The corresponding Lagrangian of optimization (29) is

$$\min_{V}\max_{\rho(s,a)\geqslant 0} L_k(V,\rho) = (1 - \gamma^{k+1})\mathbb{E}_{\mu}\left[V(s)\right] + \sum_{(s,a)\in\mathcal{S}\times\mathcal{A}}\rho(s,a)\left(\max_{\pi\in\mathcal{P}}\Delta_k^{\pi}[V](s,a)\right), \tag{31}$$

where $\Delta_k^{\pi}[V](s,a) = R(s,a) + \mathbb{E}^{\pi}_{\{s_t\}_{i=1}^{k+1}|s}\left[\sum_{i=1}^{k}\gamma^i R(s_i, a_i) + \gamma^{k+1}V(s_{k+1})\right] - V(s).$

We further simplify the optimization. Since the dual variables are positive, we have

$$\min_{V}\max_{\rho(s,a)\geqslant 0, \pi\in\mathcal{P}} L_k(V,\rho) = (1 - \gamma^{k+1})\mathbb{E}_{\mu}\left[V(s)\right] + \sum_{(s,a)\in\mathcal{S}\times\mathcal{A}}\rho(s,a)\left(\Delta_k^{\pi}[V](s,a)\right). \tag{32}$$

After clarifying these properties of the optimization corresponding to the multi-step Bellman optimality equation, we are ready to prove the Theorem 3.

**Theorem 3** *The optimal policy $\pi^*$ and its corresponding value function $V^*$ is the solution to the following saddle point problem*

$$\max_{\alpha\in\mathcal{P}(\mathcal{S}),\pi\in\mathcal{P}(\mathcal{A})}\min_V L_k(V,\alpha,\pi) \quad := \quad (1-\gamma^{k+1})\mathbb{E}_\mu\left[V(s)\right] \tag{8}$$

$$+ \sum_{\{s_i,a_i\}_{i=0}^k,s_{k+1}} \alpha(s_0)\prod_{i=0}^k \pi(a_i|s_i)p(s_{i+1}|s_i,a_i)\delta[V]\left(\{s_i,a_i\}_{i=0}^k,s_{k+1}\right)$$

*where $\delta[V]\left(\{s_i,a_i\}_{i=0}^k,s_{k+1}\right) = \sum_{i=0}^k \gamma^i R(s_i,a_i) + \gamma^{k+1}V(s_{k+1}) - V(s)$.*

**Proof** By Theorem 1 in multi-step setting, we can decompose $\rho(s,a) = \alpha(s)\pi(a|s)$ without any loss. Plugging such decomposition into the Lagrangian 32 and realizing the equivalence among the optimal policies, we arrive the optimization as $\min_V \max_{\alpha\in\mathcal{P}(\mathcal{S}),\pi\in\mathcal{P}(\mathcal{A})} L_k(V,\alpha,\pi)$. Then, because of the strong duality as we proved in Lemma 9, we can switch $\min$ and $\max$ operators in optimization 8 without any loss. ∎

**Lemma 9** *The strong duality holds in optimization (8).*

**Proof** Specifically, for every $\alpha\in\mathcal{P}(\mathcal{S}),\pi\in\mathcal{P}(\mathcal{A})$,

$$\ell(\alpha,\pi) = \min_V L_k(V,\alpha,\pi) \leqslant \min_V\left\{L_k(V,\alpha,\pi);\ \delta[V]\left(\{s_i,a_i\}_{i=0}^k,s_{k+1}\right)\leqslant 0\right\}$$

$$\leqslant \min_V\left\{\begin{array}{l}(1-\gamma^{k+1})\mathbb{E}_{s\sim\mu(s)}\left[V(s)\right],\\ \text{s.t. } \delta[V]\left(\{s_i,a_i\}_{i=0}^k,s_{k+1}\right)\leqslant 0\end{array}\right\} = (1-\gamma^{k+1})\mathbb{E}_{s\sim\mu(s)}\left[V^*(s)\right].$$

On the other hand, since $L_k(V,\alpha^*,\pi^*)$ is convex w.r.t. $V$, we have $V^* \in \text{argmin}_V L_k(V,\alpha^*,\pi^*)$, by checking the first-order optimality. Therefore, we have

$$\max_{\alpha\in\mathcal{P}(\mathcal{S}),\pi\in\mathcal{P}(\mathcal{A})}\ell(\alpha,\pi) = \max_{\alpha\in\mathcal{P}(\mathcal{S}),\pi\in\mathcal{P}(\mathcal{A}),V\in\text{argmin}_V L_k(V,\alpha,\pi)} L_k(V,\alpha,\pi)$$

$$\geqslant L(V^*,\alpha^*,\pi^*) = (1-\gamma^{k+1})\mathbb{E}_{s\sim\mu(s)}\left[V^*(s)\right].$$

Combine these two conditions, we achieve the strong duality even without convex-concave property

$$(1-\gamma^{k+1})\mathbb{E}_{s\sim\mu(s)}\left[V^*(s)\right] \leqslant \max_{\alpha\in\mathcal{P}(\mathcal{S}),\pi\in\mathcal{P}(\mathcal{A})}\ell(\alpha,\pi) \leqslant (1-\gamma^{k+1})\mathbb{E}_{s\sim\mu(s)}\left[V^*(s)\right].$$

∎

### B.2 The Composition in Applying Augmented Lagrangian Method

We consider the one-step Lagrangian duality first. Following the vanilla augmented Lagrangian method, one can achieve the dual function as

$$\ell(\alpha,\pi) = \min_V (1-\gamma)\mathbb{E}_{s\sim\mu(s)}\left[V(s)\right] + \sum_{(s,a)\in\mathcal{S}\times\mathcal{A}} P_c\left(\Delta[V](s,a),\alpha(s)\pi(a|s)\right),$$

where

$$P_c\left(\Delta[V](s,a),\alpha(s)\pi(a|s)\right) = \frac{1}{2c}\left\{\left[\max\left(0,\alpha(s)\pi(a|s)+c\Delta[V](s,a)\right)\right]^2 - \alpha^2(s)\pi^2(a|s)\right\}.$$

The computation of $P_c$ is in general intractable due to the composition of $\max$ and the condition expectation in $\Delta[V](s,a)$, which makes the optimization for augmented Lagrangian method difficult.

For the multi-step Lagrangian duality, the objective will become even more difficult due to constraints are on distribution family $\mathcal{P}(\mathcal{S})$ and $\mathcal{P}(\mathcal{A})$, rather than $\mathcal{S}\times\mathcal{A}$.

### B.3 Path Regularization

**Theorem 4** *The local duality holds for $L_r(V,\alpha,\pi)$. Denote $(V^*,\alpha^*,\pi^*)$ as the solution to Bellman optimality equation, with some appropriate $\eta_V$, $(V^*,\alpha^*,\pi^*) = \text{argmax}_{\alpha\in\mathcal{P}(\mathcal{S}),\pi\in\mathcal{P}(\mathcal{A})} \text{argmin}_V L_r(V,\alpha,\pi)$.*

**Proof** The local duality can be verified by checking the Hessian of $L_r(\theta_{V^*})$. We apply the local duality theorem (Luenberger & Ye, 2015)[Chapter 14]. Suppose $(\tilde{V}^*, \tilde{\alpha}^*, \tilde{\pi}^*)$ is a local solution to $\min_V \max_{\alpha \in \mathcal{P}(\mathcal{S}), \pi \in \mathcal{P}(\mathcal{A})} L_r(V, \alpha, \pi)$, then, $\max_{\alpha \in \mathcal{P}(\mathcal{S}), \pi \in \mathcal{P}(\mathcal{A})} \min_V L_r(V, \alpha, \pi)$ has a local solution $\tilde{V}^*$ with corresponding $\tilde{\alpha}^*, \tilde{\pi}^*$.

Next, we show that with some appropriate $\eta_V$, the path regularization does not change the optimum. Let $U^\pi(s) = \mathbb{E}^\pi \left[ \sum_{i=0}^\infty \gamma^i R(s_i, a_i) | s \right]$, and thus, $U^{\pi^*} = V^*$. We first show that for $\forall \pi_b \in \mathcal{P}(\mathcal{A})$, we have

$$\mathbb{E}\left[ \left( \mathbb{E}^{\pi_b} \left[ \sum_{i=0}^\infty \gamma^i R(s_i, a_i) \right] - V^*(s) \right)^2 \right] = \mathbb{E}\left[ \left( U_b^\pi(s) - U^{\pi^*}(s) + U^{\pi^*}(s) - V^*(s) \right)^2 \right]$$

$$= \mathbb{E}\left[ \left( U^{\pi_b}(s) - U^{\pi^*}(s) \right)^2 \right]$$

$$\leqslant \mathbb{E}\left[ \left( \int \left( \prod_{i=0}^\infty \pi_b(a_i|s_i) - \prod_{i=0}^\infty \pi^*(a_i|s_i) \right) \prod_{i=0}^\infty p(s_{i+1}|s_i, a_i) \left( \sum_{i=1}^\infty \gamma^i R(s_i, a_i) \right) d\{s_i, a_i\}_{i=0}^\infty \right)^2 \right]$$

$$\leqslant \mathbb{E}\left[ \left\| \sum_{i=1}^\infty \gamma^i R(s_i, a_i) \right\|_\infty^2 \left\| \left( \prod_{i=0}^\infty \pi_b(a_i|s_i) - \prod_{i=0}^\infty \pi^*(a_i|s_i) \right) \prod_{i=0}^\infty p(s_{i+1}|s_i, a_i) \right\|_1^2 \right]$$

$$\leqslant 4 \left\| \sum_{i=1}^\infty \gamma^i R(s_i, a_i) \right\|_\infty^2 \leqslant \frac{4}{(1-\gamma)^2} \| R(s, a) \|_\infty^2$$

where the last second inequality comes from the fact that $\pi_b(a_i|s_i)p(s_{i+1}|s_i, a_i)$ is distribution.

We then rewrite the optimization $\min_V \max_{\alpha \in \mathcal{P}(\mathcal{S}), \pi \in \mathcal{P}(\mathcal{A})} L_r(V, \alpha, \pi)$ as

$$\min_V \max_{\alpha \in \mathcal{P}(\mathcal{S}), \pi \in \mathcal{P}(\mathcal{A})} L_k(V, \alpha, \pi)$$
$$\text{s.t.} \quad V \in \Omega_{\epsilon, \pi_b} := \left\{ V : \mathbb{E}_{s \sim \mu(s)} \left[ \left( \mathbb{E}^{\pi_b} \left[ \sum_{i=0}^\infty \gamma^i R(s_i, a_i) \right] - V(s) \right)^2 \right] \leqslant \epsilon \right\},$$

due to the well-known one-to-one correspondence between regularization $\eta_V$ and $\epsilon$ Nesterov (2005). If we set $\eta_V$ with appropriate value so that its corresponding $\epsilon(\eta_V) \geqslant \frac{2}{1-\gamma} \| R(s, a) \|_\infty$, we will have $V^* \in \Omega_{\epsilon(\eta_V)}$, which means adding such constraint, or equivalently, adding the path regularization, does not affect the optimality. Combine with the local duality, we achieve the conclusion.

∎

In fact, based on the proof, the closer $\pi_b$ to $\pi^*$ is, the smaller $\mathbb{E}_{s \sim \mu(s)} \left[ \left( \mathbb{E}^{\pi_b} \left[ \sum_{i=0}^\infty \gamma^i R(s_i, a_i) \right] - V^*(s) \right)^2 \right]$ will be. Therefore, we can set $\eta_V$ bigger for better local convexity, which resulting faster convergence.

### B.4 Stochastic Dual Ascent Update

**Corollary 5** *The regularized dual function $\ell_r(\alpha, \pi)$ has gradients estimators*

$$\nabla_{\theta_\alpha} \ell_r(\theta_\alpha, \theta_\pi) = \mathbb{E}_\alpha^\pi \left[ \delta \left( \{s_i, a_i\}_{i=0}^k, s_{k+1} \right) \nabla_{\theta_\alpha} \log \alpha(s) \right],$$
$$\nabla_{\theta_\pi} \ell_r(\theta_\alpha, \theta_\pi) = \mathbb{E}_\alpha^\pi \left[ \delta \left( \{s_i, a_i\}_{i=0}^k, s_{k+1} \right) \sum_{i=0}^k \nabla_{\theta_\pi} \log \pi(a|s) \right].$$

**Proof** We mainly focus on deriving $\nabla_{\theta_\pi} \ell_r(\theta_\alpha, \theta_\pi)$. The derivation of $\nabla_{\theta_\alpha} \ell_r(\theta_\alpha, \theta_\pi)$ is similar.

By chain rule, we have

$$\nabla_{\theta_\pi} \ell_r(\theta_\alpha, \theta_\pi) = \underbrace{\left( \nabla_V L_k(V(\alpha, \theta), \alpha, \theta) - 2\eta_V \left( \mathbb{E}^{\pi_b} \left[ \sum_{i=0}^\infty \gamma^i R(s_i, a_i) \right] - V^*(s) \right) \right) \nabla_{\theta_\pi} V(\alpha, \theta)}_{0}$$

$$+ \mathbb{E}_\alpha^\pi \left[ \delta \left( \{s_i, a_i\}_{i=0}^k, s_{k+1} \right) \sum_{i=0}^k \nabla_{\theta_\pi} \log \pi(a|s) \right]$$

$$= \mathbb{E}_\alpha^\pi \left[ \delta \left( \{s_i, a_i\}_{i=0}^k, s_{k+1} \right) \sum_{i=0}^k \nabla_{\theta_\pi} \log \pi(a|s) \right].$$

The first term in RHS equals to zero due to the first-order optimality condition for $V(\alpha, \pi) = \operatorname{argmin}_V L_r(V, \alpha, \pi)$. ∎

## B.5 PRACTICAL ALGORITHM

**Theorem 6** *In $t$-th iteration, given $V^t$ and $\pi^{t-1}$,*

$$\operatorname*{argmax}_{\alpha \geqslant 0} \mathbb{E}_{\mu(s)\pi^{t-1}(s)} \left[ (\tilde{\alpha}(s) + \eta_\mu) \, \delta \left( \{s_i, a_i\}_{i=0}^k, s_{k+1} \right) \right] - \eta_\alpha \left\| \tilde{\alpha} \right\|_\mu^2$$

$$= \frac{1}{\eta_\alpha} \max \left( 0, \mathbb{E}^{\pi^{t-1}} \left[ \delta \left( \{s_i, a_i\}_{i=0}^k, s_{k+1} \right) \right] \right).$$

**Proof** Recall the optimization w.r.t. $\tilde{\alpha}$ is $\max_{\tilde{\alpha} \geqslant 0} \mathbb{E}_\mu \left[ \tilde{\alpha}(s) \mathbb{E}^\pi \left[ \delta \left( \{s_i, a_i\}_{i=0}^k, s_{k+1} \right) \right] - \eta_\alpha \tilde{\alpha}^2(s) \right]$,
denote $\tau(s)$ as the dual variables of the optimization, we have the KKT condition as

$$\begin{cases} \eta_\alpha \tilde{\alpha} & = \tau + \mathbb{E}^\pi \left[ \delta \left( \{s_i, a_i\}_{i=0}^k, s_{k+1} \right) \right], \\ \tau(s)\tilde{\alpha}(s) & = 0, \\ \tilde{\alpha} & \geqslant 0, \\ \tau & \geqslant 0, \end{cases}$$

$$\Rightarrow \begin{cases} \tilde{\alpha} & = \frac{\tau + \mathbb{E}^\pi \left[ \delta \left( \{s_i, a_i\}_{i=0}^k, s_{k+1} \right) \right]}{\eta_\alpha}, \\ \tau(s) \left( \tau(s) + \mathbb{E}^\pi \left[ \delta \left( \{s_i, a_i\}_{i=0}^k, s_{k+1} \right) \right] \right) & = 0, \\ \tilde{\alpha} & \geqslant 0, \\ \tau & \geqslant 0, \end{cases}$$

$$\Rightarrow \tau(s) = \begin{cases} -\mathbb{E}^\pi \left[ \delta \left( \{s_i, a_i\}_{i=0}^k, s_{k+1} \right) \right] & \mathbb{E}^\pi \left[ \delta \left( \{s_i, a_i\}_{i=0}^k, s_{k+1} \right) \right] < 0 \\ 0 & \mathbb{E}^\pi \left[ \delta \left( \{s_i, a_i\}_{i=0}^k, s_{k+1} \right) \right] \geqslant 0 \end{cases}.$$

Therefore, in $t$-th iteration, $\tilde{\alpha}^t(s) = \frac{1}{\eta_\alpha} \max \left( 0, \mathbb{E}^\pi \left[ \delta \left( \{s_i, a_i\}_{i=0}^k, s_{k+1} \right) \right] \right)$. ∎

## C EXPERIMENT DETAILS

**Policy and value function parametrization.** For fairness, we use the same parametrization across all the algorithms. The parametrization of policy and value functions are largely based on the recent paper by Rajeswaran et al. (2017), which shows the natural policy gradient with the RBF neural network achieves the state-of-the-art performances of TRPO on MuJoCo. For the policy distribution, we parametrize it as $\pi_{\theta_\pi}(a|s) = \mathcal{N}(\mu_{\theta_\pi}(s), \Sigma_{\theta_\pi})$, where $\mu_{\theta_\pi}(s)$ is a two-layer neural nets with the random features of RBF kernel as the hidden layer and the $\Sigma_{\theta_\pi}$ is a diagonal matrix. The RBF kernel bandwidth is chosen via median trick (Dai et al., 2014; Rajeswaran et al., 2017). The same as Rajeswaran et al. (2017), we use 100 hidden nodes in Pendulum, InvertedDoublePendulum, Swimmer, Hopper, and use 500 hidden nodes in HalfCheetah and Walker. Since the TRPO and PPO uses GAE (Schulman et al., 2015b) with linear baseline as $V$, we also use the parametrization for $V$ in our algorithm. However, the Dual-AC can adopt arbitrary function approximator without any change.

**Training details.** We report the hyperparameters for each algorithms here. We use the $\gamma = 0.995$ for all the algorithms. We keep constant stepsize and tuned for TRPO, PPO and Dual-AC in $\{0.001, 0.01, 0.1\}$. The batchsize are set to be 52 trajectories for comparison to the competitors in Section 6.2. For the Ablation study, we set batchsize to be 24 trajectories for faster runtime. The CG damping parameter for TRPO is set to be $10^{-4}$. We iterate 20 steps for the Fisher information matrix computation. For the $\eta_V, \eta_\mu, \frac{1}{\eta_\alpha}$ in Dual-AC from $\{0.001, 0.01, 0.1, 1\}$.

