# OpenReview forum: "Boosting the Actor with Dual Critic"
_ICLR.cc/2018/Conference — Accept (Poster)_

### Official Review · AnonReviewer1 · 2017-11-27
**an interesting paper**

**Rating:** 7
**Confidence:** 4

**Review:**

This paper studies a new architecture DualAC. The author give strong and convincing justifications based on the Lagrangian dual of the Bellman equation (although not new, introducing this as the justification for the architecture design is plausible).

There are several drawbacks of the current format of the paper:
1. The algorithm is vague. Alg 1 line 5: 'closed form': there is no closed form in Eq(14). It is just an MC approximation.
line 6: Decay O(1/t^\beta). This is indeed vague albeit easy to understand. The algorithm requires that every step is crystal clear.

2. Also, there are several format error which may be due to compiling, e.g., line 2 of Abstract,'Dual-AC ' (an extra space). There are many format errors like this throughout the paper. The author is suggested to do a careful format check.

3. The author is suggested to explain more about the necessity of introducing path regularization and SDA. The current justification is reasonable but too brief.

4. The experimental part is ok to me, but not very impressive.

Overall, this seems to be a nice paper to me.

---

> ### Author Response · Authors · 2018-01-03
> **Response to Reviewer1**
>
> Thanks for the constructive suggestions.
>
> We modified the stepsize decay form more concretely (line 6 of Alg 1). It is adjusted based on the theoretical requirement for convergence [2, 3]
>
> We fixed the extra space after `Dual-AC'.
>
> We added more discussion of the benefits and the necessity of the path-regularization and stochastic dual ascent in the updated version in the 2nd paragraph and 3rd paragraph in page 5, respectively. For better illustrating the necessity of path-regularization and stochastic dual ascent, we also added more empirical experiments in the ablation study part in Figure 1.
>
> For the experiment parts, we picked the **best** implementation of the state-of-the-art TRPO and PPO as our baselines based on the recent comprehensive comparison [1]. With the best implementations of TRPO and PPO, these two algorithms consistently achieve the best performance in most of the MuJoCo tasks, beating other alternatives, e.g., DDPG and ACKTR, with significant margins in [1].  Despite such strong baselines, our Dual-AC algorithm still shows substantial gain in 5 out of 6 domains (Fig 2), with a tie in the Swimmer-v1 task.  In InvertedDoublePendulum-v2, Dual-AC achieves almost 3x reward of TRPO and 4x of PPO.
>
> [1], Deep reinforcement learning that matters. Peter Henderson, Riashat Islam, Philip Bachman, Joelle Pineau, Doina Precup, David Meger. AAAI 2018.
> [2], Robust stochastic approximation approach to stochastic programming. A Nemirovski, A Juditsky, G Lan, A Shapiro. SIAM Journal on optimization 19 (4), 1574-1609.
> [3], Stochastic first-and zeroth-order methods for nonconvex stochastic programming. S Ghadimi, G Lan. SIAM Journal on Optimization 23 (4), 2341-2368.

---

### Official Review · AnonReviewer3 · 2017-11-27
**Good contribution**

**Rating:** 6
**Confidence:** 3

**Review:**

The paper is well written, and the authors do an admirable job of motivating their primary contributions throughout the early portions of the paper. Each extension to the Dual Actor-Critic is well motivated and clear in context. Perhaps the presentation of these extensions could be improved by providing a less formal explanation of what each does in practice; multi-step updates, regularized against MC returns, stochastic mirror descent.

The practical implementation section losses some of this clear organization, and could certainly be clarified each part tied into Algorithm 1, and this was itself made less high-level. But these are minor gripes overall.

Turning to the experimental section, I think the authors did a good job of evaluating their approach with the ablation study and comparisons with PPO and TRPO. There were a few things that jumped out to me that I was surprised by. The difference in performance for Dual-AC between Figure 1 and Figure 2b is significant, but the only difference seems to be a reduce batch size, is this right? This suggests a fairly significant sensitivity to this hyperparameter if so.

Reproducibility in continuous control is particularly problematic. Nonetheless, in recent work PPO and TRPO performance on the same set of tasks seem to be substantively different than what the authors get in their experiments. I'm thinking in particular of:

Proximal Policy Optimization Algorithms (Schulman et. al., 2017)
Multi-Batch Experience Replay for Fast Convergence of Continuous Action Control (Han and Sung, 2017)

In both these cases the results for PPO and TRPO vary pretty significantly from what we see here, and an important one to look at is the InvertedDoublePendulum-v1 task, which I would think PPO would get closer to 8000, and TRPO not get off the ground. Part of this could be the notion of an "iteration", which was not clear to me how this corresponded to actual time steps. Most likely, to my mind, is that the parameterization used (discussed in the appendix) is improving TRPO and hurting PPO.

With these in mind I view the comparison results with a bit of uncertainty about the exact amount of gain being achieved, which may beg the question if the algorithmic contributions are buying much for their added complexity?

Pros:
Well written, thorough treatment of the approaches
Improvements on top of Dual-AC with ablation study show improvement

Cons:
Empirical gains might not be very large

---

> ### Author Response · Authors · 2018-01-03
> **Response to Reviewer3**
>
> Thanks for the constructive comments.
>
> As suggested by the reviewer, we provided further details to explain the benefits of several important extensions: path regularization (2nd paragraph on page 5), stochastic dual ascent (1st paragraph of section 4.3 on page 5), practical updates for policy (the paragraphs surrounding Eqns 16 & 17 on page 7) and critic (2nd last paragraph on page 6).
>
> The gaps between Figs 1 and 2 are indeed mainly due to the batch size used in the algorithm. As expected, the batch size affects the variance of the gradients, thereby affecting the convergence of the algorithm.  Such an effect is not unique to our algorithm and has been observed in the literature; see for example similar results for the TRPO baseline in a recent empirical study [1].
>
> For the comparison between the TRPO and PPO, the recent empirical study [1] shows that different implementations will affect their performance a lot.  Based on the evaluation results in [1], we compared our algorithm with the **best** implementation of TRPO, i.e., the original implementation by Schulman, 2015. From Table 1 and Figure 26 in [1], we can see that the best implementation of TRPO may achieve comparable or even better results comparing to PPO on several tasks.  On the other hand, we used the same parametrization for all the algorithms, which may be preferable to TRPO. We follow [2] using the “iteration” in the experiments to illustrate  the policy behaviors along with the number of updates in the algorithm, rather than the number of data collected for a better understanding of the algorithms in terms of each update.
>
> Re gains of our algorithm: Since the major contribution of our paper is a new algorithm, rather than an alternative parametrization, we conduct the comparison with the baseline using the same parametrizations for fairness. We did not introduce any extra complexity in terms of parameterization. In terms of updates in algorithm, although the update rule for value function needs an extra sample reweighting,  the update rule for policy is much simpler than TRPO, which requires extra adjustments for policy and related parameters.  Therefore, the gains are **not** achieved by added complexity.
>
> [1], Deep reinforcement learning that matters. Peter Henderson, Riashat Islam, Philip Bachman, Joelle Pineau, Doina Precup, David Meger, AAAI 2018.
> [2], Towards generalization and simplicity in continuous control. Aravind Rajeswaran, Kendall Lowrey, Emanuel Todorov, Sham Kakade, NIPS 2017.

---

### Official Review · AnonReviewer2 · 2017-11-27
**Overall a good paper, with a few details that need clarification**

**Rating:** 5
**Confidence:** 4

**Review:**

This paper proposes a method, Dual-AC, for optimizing the actor(policy) and critic(value function) simultaneously which takes the form of a zero-sum game resulting in a principled method for using the critic to optimize the actor. In order to achieve that, they take the linear programming approach of solving the bellman optimality equations, outline the deficiencies of this approach, and propose solutions to mitigate those problems. The discussion on the deficiencies of the naive LP approach is mostly well done. Their main contribution is extending the single step LP formulation to a multi-step dual form that reduces the bias and makes the connection between policy and value function optimization much clearer without loosing convexity by applying a regularization. They perform an empirical study in the Inverted Double Pendulum domain to conclude that their extended algorithm outperforms the naive linear programming approach without the improvements. Lastly, there are empirical experiments done to conclude the superior performance of Dual-AC in contrast to other actor-critic algorithms.

Overall, this paper could be a significant algorithmic contribution, with the caveat for some clarifications on the theory and experiments. Given these clarifications in an author response, I would be willing to increase the score.

For the theory, there are a few steps that need clarification and further clarification on novelty. For novelty, it is unclear if Theorem 2 and Theorem 3 are both being stated as novel results. It looks like Theorem 2 has already been shown in "Randomized Linear Programming Solves the Discounted Markov Decision Problem in Nearly-Linear Running Time”. There is a statement that “Chen & Wang (2016); Wang (2017) apply stochastic first-order algorithms (Nemirovski et al., 2009) for the one-step Lagrangian of the LP problem in reinforcement learning setting. However, as we discussed in Section 3, their algorithm is restricted to tabular parametrization”. Is you Theorem 2 somehow an extension? Is Theorem 3 completely new?

This is particularly called into question due to the lack of assumptions about the function class for value functions. It seems like the value function is required to be able to represent the true value function, which can be almost as restrictive as requiring tabular parameterizations (which can represent the true value function). This assumption seems to be used right at the bottom of Page 17, where U^{pi*} = V^*. Further, eta_v must be chosen to ensure that it does not affect (constrain) the optimal solution, which implies it might need to be very small. More about conditions on eta_v would be illuminating.

There is also one step in the theorem that I cannot verify. On Page 18, how is the squared removed for difference between U and Upi? The transition from the second line of the proof to the third line is not clear. It would also be good to more clearly state on page 14 how you get the first inequality, for || V^* ||_{2,mu}^2.


For the experiments, the following should be addressed.

1. It would have been better to also show the performance graphs with and without the improvements for multiple domains.

2. The central contribution is extending the single step LP to a multi-step formulation. It would be beneficial to empirically demonstrate how increasing k (the multi-step parameter) affects the performance gains.

3. Increasing k also comes at a computational cost. I would like to see some discussions on this and how long dual-AC takes to converge in comparison to the other algorithms tested (PPO and TRPO).

4. The authors concluded the presence of local convexity based on hessian inspection due to the use of path regularization. It was also mentioned that increasing the regularization parameter size increases the convergence rate. Empirically, how does changing the regularization parameter affect the performance in terms of reward maximization? In the experimental section of the appendix, it is mentioned that multiple regularization settings were tried but their performance is not mentioned. Also, for the regularization parameters that were tried, based on hessian inspection, did they all result in local convexity? A bit more discussion on these choices would be helpful.

Minor comments:
1. Page 2: In equation 5, there should not be a 'ds' in the dual variable constraint

---

> ### Author Response · Authors · 2018-01-03
> **Response to Reviewer2**
>
> We appreciate the constructive comments on both theoretical and empirical aspects by the reviewer.
>
> We first emphasize our contributions. The major contributions of this paper are (1) the **first** establishment of the competition between actor and critic in a **multi-step** setting; and (2) a novel algorithm that is make effective thanks to several critical components we introduce, including path-regularization and stochastic dual ascent, to deal with potential numerical issues that arise when one directly solves the zero-sum game.
>
> Theory Clarification:
> 1, Novelty of Theorems: Theorem 2 (one-step dual form)  is indeed an extension of existing results to continuous state and action MDP.  Theorem 3 (multi-step dual form)  is one of our major contributions and is a novel result.  The claim in Theorem 3 may appear natural, but its proof is a highly nontrivial generalization of the one-step case, since the convex-concave structure breaks down in the multi-step setting.  We have made this clearer in the revision.
>
> 2, Assumptions on value function class and choice of regularization parameter:  We tried to separate the justification of path-regularization (theory) from the parametrization of value function (practice).
> 	i), Theoretically, regarding Theorem 4 and its proof, we consider the entire value function space, i.e., the nonparametric limit, without taking into account of parametrization. Hence, as long as the regularization parameter (i.e., eta) is selected appropriately, this doesn’t affect the optimality. Note that an implicit condition of eta is provided on Page 18; however, finding an explicit condition for the regularization parameter seems to be rather difficult and is beyond the scope of this work.
> 	ii), Practically, we always parametrize the value function (which affects the valid range of eta) and tune the regularization parameter to achieve the best performance.
>
> 3, Minor gaps in proofs:  Yes, there should be a square in the proof on page 18. This does not jeopardize the rest of the proof as we only need boundedness of this term.  We have fixed the issue in our revision. For the first inequality on page 14 about ||V^*||_{2, .mu}^2, it comes from the inequality E[(X+Y)^2] <= 2 * (E[X^2] + E[Y^2]), a generalization of (a+b)^2<= 2(a^2 +b^2).  We have added more details to the proof. Thanks for pointing out these issues.
>
>
> Experiments Clarification:
> 1, Performance comparisons with and without the improvements:  In the ablation experiment part, we compared the proposed dual-AC algorithm with/without the path-regularization, and with/without multi-step on several MuJoCo tasks, including InvertedDoublePendulum, Swimmer, and Hopper. The results suggest that using path-regularization and multi-step significantly improves the performances.   Detailed experimental results can be found in Figure 1.
>
> 2, Effects of the length of multi-steps: We conducted additional experiments to investigate the effects of multi-step lengths. Specifically, we compared the performance with different k = {1, 10, 50}, and tested on three tasks. Better performances are observed with increasing k, which indicates that reducing the bias is indeed critical. Detailed experimental results can be found in Figure 1.
>
> 3, Computation overheads of using multi-step: In terms of the computational cost, assuming the length of the trajectories are m, with a simple moving sum algorithm, we calculate all the k length partial reward sums for a trajectory in O(m) with a O(1) amortized cost to calculate each sum of rewards. This method was used in our experiments for our algorithm as well as all competitors. Comparing to TRPO and PPO, the cost for summation is the same for all algorithms and the update costs are constant for each individual algorithm regardless of the choice of k.
>
> 4, Local convexity in practice:  In general, using a positive eta_V coefficient with path-regularization always enhances the local convexity. For example, if V is parametrized in a linear form, as long as eta_V is not zero, local convexity will hold.  A larger eta_V will result in faster convergence, at the cost of extra bias. It is not easy to theoretically/empirically inspect the exact local convexity condition when a complicated parametrization of V is used. In practice, we suggest to simply tune the regularization parameter, and that’s what we have done in the experiments.

---

### Author Response · Authors · 2018-01-05
**Revised manuscript**

Thanks for the constructive reviews and comments!

We have submitted our updated manuscript with a few revisions and more experiments for clarity accordingly, including:

1, Discussion about the parametrization effect w.r.t. the path regularization.

2, More explanation on the benefits of the proposed several important extensions.

3, More details for proofs in Appendix.

4, More ablation experiments with different k = {1, 10, 50} on two more MuJoCo tasks, i.e., Swimmer-v1 and Hopper-v1, and more comparison with TRPO and PPO on Walker-v1.

---

### Decision · Program_Chairs · 2018-01-29
**ICLR 2018 Conference Acceptance Decision**

**Decision:**

Accept (Poster)

**Comment:**

All of the reviewers agree that the paper clearly presents promising ideas in developing a novel actor critic algorithm. The experiments do not show a significant gain against the baselines, but they support the presented ideas. I appreciated the ablation study on dual-AC.

Detailed comments:
My understanding is that the x-axis in Figures 1 & 2 shows the number of iterations each of which contains batch_size*1000 environment steps. It is more standard to show those plots in terms of the number of environment steps. Further, the optimal batch_size for different algorithms may be different, so using the same batch_size for all of the algorithms is not fair.